# Chemical inhomogeneity–induced profuse nanotwinning and phase transformation in AuCu nanowires

Chengpeng Yang[1,5], Bozhao Zhang[2,5], Libo Fu[1], Zhanxin Wang[1], Jiao Teng[3], Ruiwen Shao[4], Ziqi Wu[4], Xiaoxue Chang[4], Jun Ding [2]✉, Lihua Wang [1]✉ & Xiaodong Han [1]✉

Nanosized metals usually exhibit ultrahigh strength but suffer from low homogeneous plasticity. The origin of a strength–ductility trade-off has been well studied for pure metals, but not for random solid solution (RSS) alloys. How RSS alloys accommodate plasticity and whether they can achieve synergy between high strength and superplasticity has remained unresolved. Here, we show that face-centered cubic (FCC) RSS AuCu alloy nanowires (NWs) exhibit superplasticity of ~260% and ultrahigh strength of ~6 GPa, overcoming the trade-off between strength and ductility. These excellent properties originate from profuse hexagonal close-packed (HCP) phase generation (2H and 4H phases), recurrence of reversible FCC-HCP phase transition, and zigzag-like nanotwin generation, which has rarely been reported before. Such a mechanism stems from the inherent chemical inhomogeneity, which leads to widely distributed and overlapping energy barriers for the concurrent activation of multiple plasticity mechanisms. This naturally implies a similar deformation behavior for other highly concentrated solid-solution alloys with multiple principal elements, such as high/medium-entropy alloys. Our findings shed light on the effect of chemical inhomogeneity on the plastic deformation mechanism of solid-solution alloys.

Nanosized face-centered cubic (FCC) metals such as nanowires (NWs) have attracted extensive interest because of their potential applications in micro/nanoelectromechanical devices and flexible electrodes[1,2]. For practical applications, NWs are usually under stress/strain environments, in which both high strength and high ductility are desired. The strength and ductility are directly related to the atomic-scale structural evolution mechanism under strain/stress conditions[3–6]. Thus, achieving both high strength and high ductility, as well as revealing atomic-scale mechanisms, is greatly desired in metal science

and technology research, which can provide scientific guidance for designing metals with desired properties. Owing to the importance of these issues, a large number of studies have been carried out on nanosized pure metals[7–14], which have shown that nanosized metals exhibit a strength-ductility trade-off phenomenon, i.e., nanosized pure metals exhibit high strengths but suffer from low ductility at room temperature[15–18]. At relatively high temperatures, nanostructured metals can exhibit superelongation, but very low strength[19–21]. This strength-ductility trade-off phenomenon is based on studies of pure

[1]Faculty of Materials and Manufacturing, Institute of Microstructure and Property of Advanced Materials, Beijing University of Technology, Beijing 100124, China. [2]Center for Alloy Innovation and Design, State Key Laboratory for Mechanical Behavior of Materials, Xi'an Jiaotong University, Xi'an, China. [3]Department of Material Physics and Chemistry, University of Science and Technology Beijing, Beijing 100083, China. [4]Beijing Advanced Innovation Center for Intelligent Robots and Systems, School of Medical Technology, Beijing Institute of Technology, Beijing 100081, China. [5]These authors contributed equally: Chengpeng Yang, Bozhao Zhang. ✉e-mail: dingsn@xjtu.edu.cn; wlh@bjut.edu.cn; xdhan@bjut.edu.cn

metals, such as Ni, Au, and Cu NWs[12,13,22–24]. It has also been unclear how to make alloy NWs achieve both high strength and superplasticity. Based on previous classic experiments and theoretical studies conducted on pure metals, NWs exist in a low dislocation density state, high stress is required for the nucleation of new dislocations, and dislocations can easily escape rather than multiply, leading to metallic NWs exhibiting high strength but low ductility[25–27]. Many interesting studies have also shown that the plastic deformation in pure metals is dominated by partial dislocation and twinning when the size of metals is below several tens of nanometers[12,13,22,23,28,29], and full dislocations are rarely observed[17,30–32]. In a pure metal, it is easier to thicken an existing stacking fault (SF) or twin than to nucleate a new SF in the pristine lattice during plastic deformation. Molecular dynamics (MD) simulations have predicted that as the size of pure metals decreases to several nanometers, a phase transition from the FCC to body-centered cubic (BCC) phase[33–36] occurs during deformation. FCC-hexagonal close-packed (HCP) transformation is rarely observed under conventional straining, where the FCC-HCP transformation requires ultrahigh stress[37–39]. However, most proposed deformation models are based on nanosized pure metals. For highly concentrated solid-solution alloy NWs, in which the solid-solution atoms are randomly distributed, the accommodation of plastic deformation remains an unresolved issue. Thus, obtaining in situ atomic-scale deformation evidence can determine whether both high strength and superplasticity can be achieved and whether there are novel deformation mechanisms for nanosized concentrated solid-solution alloy NWs.

In this study, we present in situ atomic-scale evidence that a random solid solution (RSS) structure enables AuCu NWs to exhibit superplasticity with ~260% uniform elongation and ultrahigh strength of ~6 GPa. We discovered that the excellent properties originate from the chemical inhomogeneity (similar to that of high-entropy alloys), which leads to widely distributed and overlapping energy barriers for multiple plasticity mechanisms and enables the AuCu alloy NWs to exhibit several unconventional plastic deformation modes, such as an ultrahigh density of deformation-induced HCP lamellae (including 2H and 4H phases), zigzag-like nanotwins, and reversible FCC–HCP phase transition. These deformation modes hinder the dislocation motion, which ultimately leads to superplasticity and ultrahigh strength in the AuCu NWs.

## Results and discussion

AuCu NWs for in situ transmission electron microscopy (TEM) straining experiments were fabricated from thin films deposited on single-crystal NaCl substrates using magnetron sputtering (Fig. 1a). First, the AuCu thin film was attached to our custom-made device[40–43]. Then, ligaments generated using a Fischione 1040 NanoMill served as AuCu NWs, as shown in Fig. 1b (more details in the Methods section). Figure 1c shows the atomic-scale high-angle annular dark-field (HAADF) image, which shows the FCC structure of the AuCu alloy used in the experiments. The corresponding two-dimensional (2D) intensity mapping extracted from the HAADF image shows that the intensity is diverse at different atomic columns (see Supplementary Fig. 1), indicating that the solid-solution atoms are randomly distributed. Figure 1d, e displays the atomic-scale element mapping of the AuCu alloy; the average ratio of Au to Cu is 6:4. Interestingly, as shown in Fig. 1f, the quantified 3D mapping extracted from Fig. 1d and e (see Methods section for detailed calculation process) further indicates that the ratio of Cu is not constant, but varies in different regions (the Au mapping is shown in Supplementary Fig. 2). Specifically, in Fig. 1g, the ratio of Cu, within the range of 30% to 45%, is quantitatively denoted by colors ranging from light brown to dark brown. The varying colors at different atoms, columns, and positions confirm that the solid-solution atoms are randomly distributed in the AuCu NW.

Figure 2 displays a series of high-resolution TEM (HRTEM) images captured along the [1$\bar{1}$0] axis, which show the atomic-scale process of an RSS AuCu NW exhibiting superelongation (see details in Supplementary Fig. 3 in the Supplementary Material). To track the plastic strain, two regions with no obvious change were used as references, as indicated by the red lines. Then, the plastic strain ($\varepsilon$) was calculated using the formula $\varepsilon = (L_n - L_0)/L_0$, where $L_0$ is the initial length between

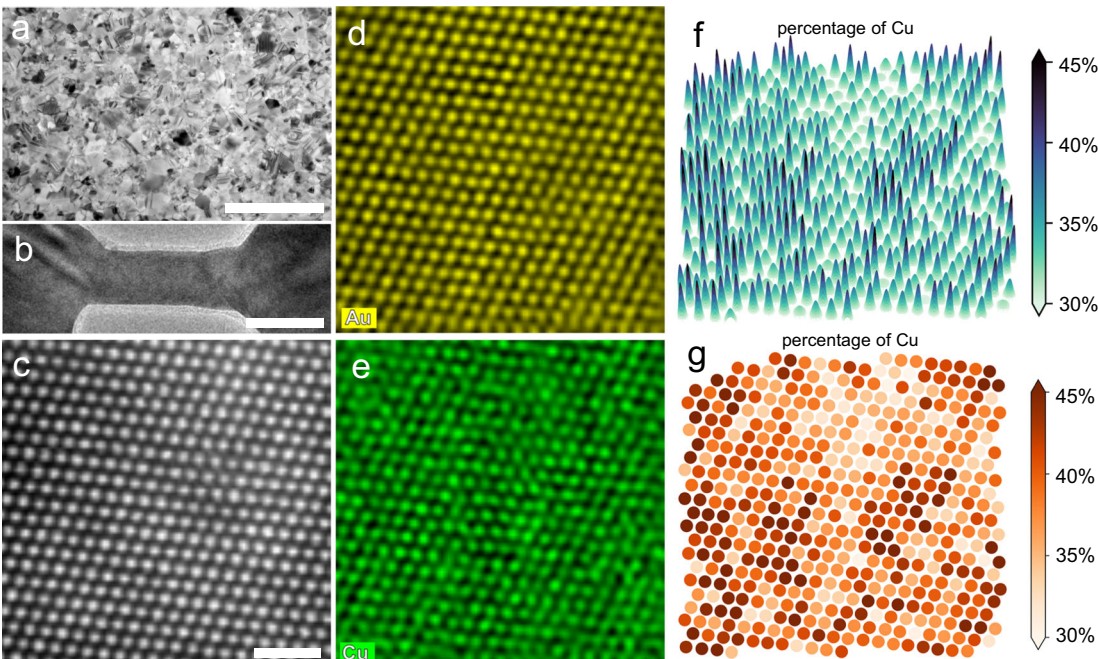

**Fig. 1 | Microstructure and Elemental distribution of AuCu nanowires (NWs).**
**a** Transmission electron microscopy (TEM) image of AuCu film used for fabricating AuCu nanowires (NWs). The scale bar is 500 nm. **b** TEM image showing a typical AuCu NW that was prepared by NanoMill. The scale bar is 25 nm. **c**–**e** High-angle annular dark-field (HAADF) image and corresponding atomic-scale element mapping of the AuCu alloy. The scale bar is 1 nm. **f**, **g** Two- and three-dimensional element mapping calculated from (**d**) and (**e**) demonstrating that the solid-solution atoms are randomly distributed in the AuCu NW.

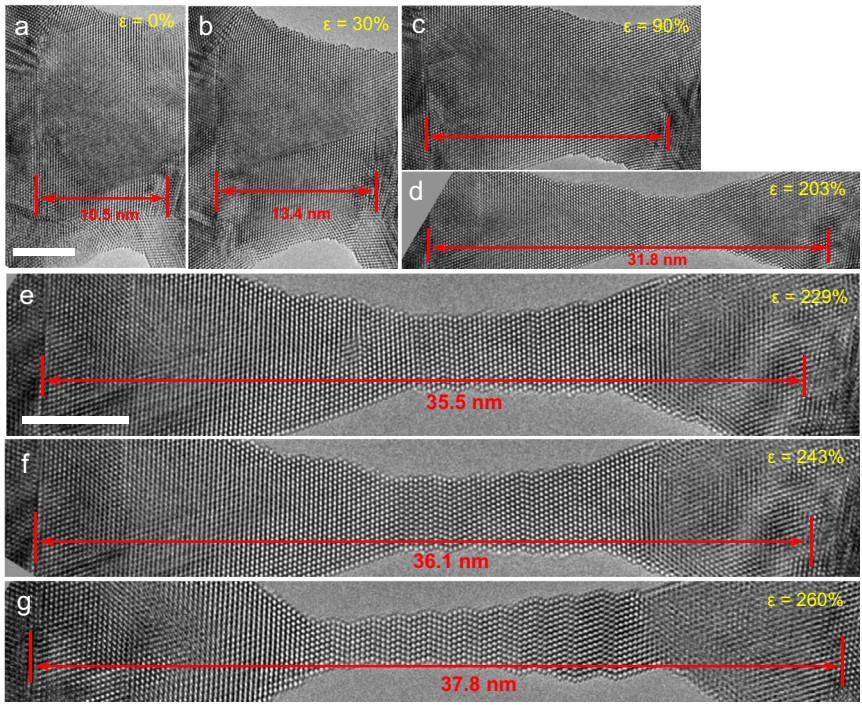

**Fig. 2 | In situ observation of super-plasticity of AuCu nanowires. a–g** A series of high-resolution transmission electron microscopy (HRTEM) images showing the superplastic elongation of the AuCu alloy NW. The initial length of the NW is ~10.5 nm. Upon deformation, the NW first experienced elastic elongation and then was subjected to a homogeneous elongation as large as ~260%. The references are indicated by the short red line; ε represents the strain. The scale bar is 5 nm.

the two reference points, and $L_n$ is the length after deformation. As shown in Fig. 1a, for an AuCu NW with [111] orientation, the initial length of the NW is ~10.5 nm ($L_0$). Upon deformation, the NW first experienced elastic elongation and then was subjected to a homogeneous elongation as large as ~260%, as shown in Fig. 2a–g. Comparison of the atomic-scale structure of the NW in Fig. 2a with those in Fig. 2b–g shows that the NW exhibited a dual-stage plastic deformation that led to superplasticity. In Stage 1, the plastic deformation was mediated by extended dislocations and full dislocations that resulted from a leading partial dislocation followed by a trailing partial dislocation (see examples in Supplementary Figs. 4 and 5). These dislocations quickly escaped from the small NW with no debris, resulting in the nearly defect-free AuCu NW during loading (as shown in Figs. 2c, d and 4). Such dislocation activities on multiple slip systems lead to the RSS AuCu NW exhibiting superplasticity, which was frequently observed in our experiments[41]. We also performed the tensile experiments of AuCu NWs under beam blank (only turning on the electron beam for image capture), while the super-elongation was still observed (as shown in Supplementary Fig. 6), indicating that the electron beam irradiation had no significant impact on the exceptional ductility of AuCu NWs. In Stage 2, the NW accommodated plastic strain via the FCC–HCP phase transformation and multiple sub-nanosized twin formations, leading to the RSS AuCu NW exhibiting a zigzag-like structure with an arrangement of (111) coherent twin boundaries (Fig. 2e–g) that endowed the AuCu NW with ultrahigh strength. These deformation modes were considered to trigger the superplasticity and ultrahigh strength of this RSS AuCu NW.

Figure 3a–c displays enlarged HRTEM images of the dual-stage plastic deformation. To quantify the stress values at different deformation stages, we evaluated the elastic strain (along the [111] direction) during the deformation by comparing the atomic lattice distance of the strained NWs with that of the unstrained region (see details in Supplementary Fig. 7). To evaluate the elastic strain more accurately, different methods were used to measure the elastic strain; the results show that the standard deviation is below 0.3%, and similar

elastic strain values were obtained using different methods (as shown in Supplementary Figs. 8 and 9). This indicates that the elastic strain values are relatively accurate and reliable. The stress values were then estimated by multiplying Young's modulus by the elastic strain. Figure 3d shows the stress–strain curve of the AuCu NWs, showing a dual-stage response feature. During loading, the NW experienced elastic strain, followed by yielding with a yielding stress of ~2 GPa (Young's modulus of 115.18 GPa in the AuCu alloy[44]). The NW then underwent Stage 1, in which the plasticity was governed by leading and trailing partial dislocations that formed extended dislocations or full dislocations on multiple gliding systems (as shown in Fig. 3a, Supplementary Figs. 4 and 5). During this process, the uniform elongation of the NW approached ~200%, and the corresponding stress increased gradually and then plateaued at ~3.5 GPa (Fig. 3d). This highly active dislocation-induced superplasticity with ultrahigh strength was also observed in other RSS AuCu NWs[41]. In Stage 2, the plastic deformation was governed by the FCC–HCP phase transition and nanotwin formation, leading to the NW containing a high density of phase boundaries and (111) coherent twin boundaries (as shown in Fig. 3b, c). At this stage, the strength of the NW increased gradually and approached ~6 GPa with a uniform elongation of ~260% (Fig. 3d). Figure 3e shows the statistical data of the elongation versus the strength of various metallic materials, including metallic NWs, high-entropy alloys, gradient nanostructured metals, HCP metals, FCC and BCC metals, steels, Cu/Cu alloys, and Al/Al alloys[4,12,14–17,23,32,45–74]; the star denotes the data of the RSS AuCu NW in our experiment. Most of the metals exhibited a trade-off between strength and ductility, whereas the RSS AuCu NW exhibited superplasticity with ultrahigh strength, overcoming the strength–ductility trade-off. In addition, we conducted tensile experiments on Au NWs (as indicated in Fig. 3e), which exhibited a homogeneous plasticity of ~85% and strength of ~2.3 GPa (also see Supplementary Fig. 10). These results are similar to previously reported experimental studies on Au[45]. Comparing Fig. 2 with Supplementary Fig. 10, one can see that the RSS AuCu alloy NWs indeed overcome the strength–ductility trade-off.

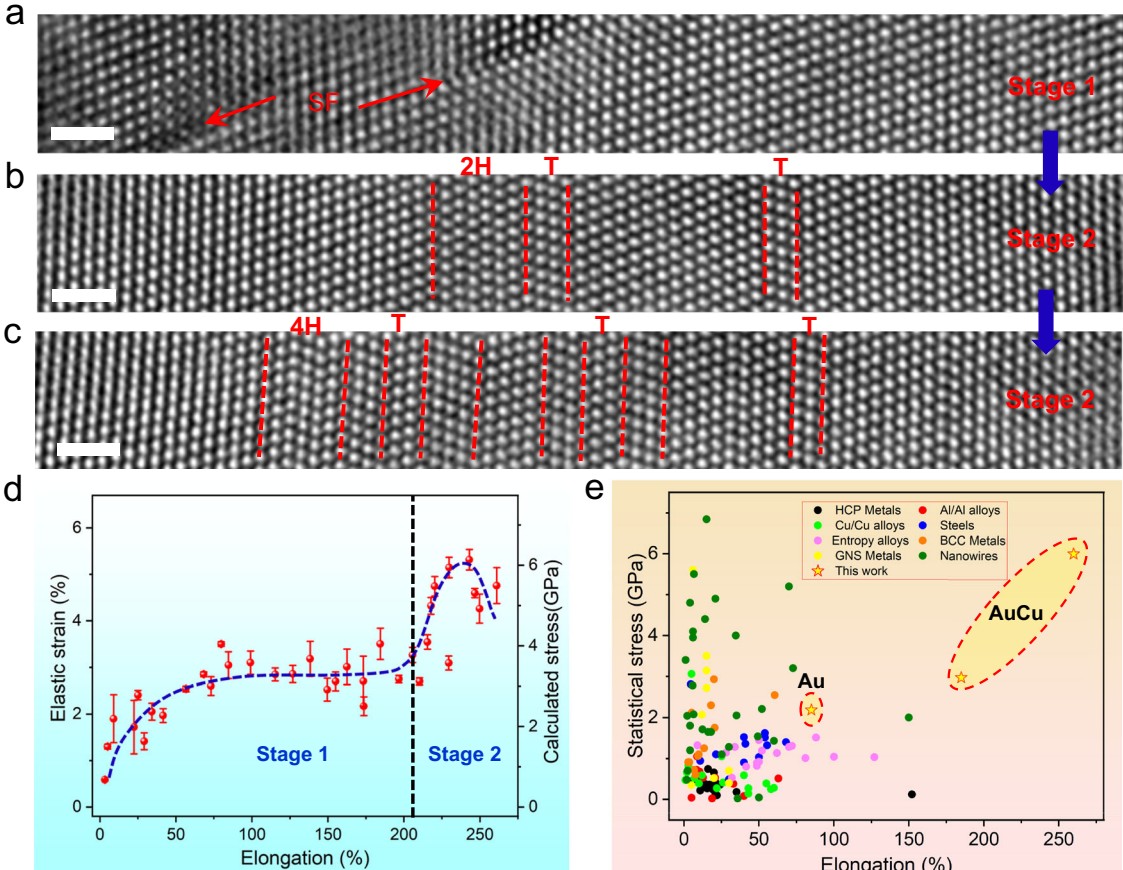

**Fig. 3 | Microstructure and mechanical properties of AuCu NWs. a–c** Three enlarged HRTEM images captured from different deformation processes. "SF", "2H, 4H", and "T" represent stacking fault, hexagonal phase, and twin, respectively. The scale bar is 1 nm. **d** Statistical data of the elastic strain as a function of the uniform elongation. The elastic strain/stress can be divided into Stage 1, corresponding to Fig. 2a, and Stage 2, corresponding to Fig. 2b, c. The error bar represents the standard deviation. The blue dot line is the corresponding fitting curve. **e** Comparison of the mechanical properties of the AuCu alloy NWs with those of other high-performing materials. The star represents the results of this study. The statistical data are divided into nine categories: HCP metals (black), Cu/Cu alloys (green), metal nanowires (dark yellow), entropy alloys (magenta), GNS metals (yellow), Al/Al alloys (red), steels (blue), BCC metals (olive), and this work (stars).

Figure 4 displays typical in situ atomic-scale images captured during Stage 1, which show the generation and annihilation of SFs that resulted from leading followed by trailing partial dislocations (generating extended dislocations or full dislocations, as shown in Supplementary Figs. 4 and 5). As shown in Fig. 4a, b, two SFs (denoted SF1 and SF2) were generated in the NW, resulting from two partial dislocations with a Burgers vector of $\mathbf{b} = 1/6[112]$. As the strain increased, a trailing partial dislocation was generated on the same glide plane of SF2 and erased the pre-existing SF2, as shown in Fig. 4c. As shown in Fig. 4d, SF1 was annihilated by the same mechanism as SF2, leading to this region being defect-free (Fig. 4d). With continued straining, a new partial (denoted SF3) and full dislocation (marked with ⊥) was generated (Fig. 4e). With further loading, SF3 was also erased owing to the generation of subsequent partial dislocations, as shown in Fig. 4f. This leading partial dislocation followed by trailing partial dislocation on multiple slip systems can effectively alleviate the deformation localization, thereby enabling the RSS AuCu NW to exhibit superplasticity via sequential nucleation, motion, and annihilation of dislocations. This dislocation behavior is different from that observed in many previous studies, which reported that the plasticity of small-sized pure metals is governed by partial dislocation and twinning[12,13,22,23,28,29].

Figure 5 displays a series of atomic-scale images captured along the [1̄10] axis as the strain increased from ~200% to ~250%, in which the plasticity of the NW was governed by the FCC–HCP phase transformation and multiple sub-nanosized twin formations. Figure 5a shows

two SFs that resulted from partial dislocations with $\mathbf{b} = 1/6[11\bar{2}]$ on the (111) slip plane (denoted as SF1 and SF2, marked by the red arrow). During this process, SF1 is stable with no obvious change and thus serves as a reference. Upon deformation, a partial dislocation was emitted on the adjacent (111) plane of SF2, leading to a two-atomic-layered twin formation, marked as T1 in Fig. 5b. Simultaneously, a three-atomic-layered twin, marked as T2, was also observed. Figure 5c, d presents a full dislocation (denoted T) and an SF (denoted SF3) resulting from partial dislocation as the strain increased. With continued straining, the full dislocation was annihilated, and a phase transition from the FCC structure to an HCP structure with an AB stacking sequence (2H phase) was directly observed, as shown in Fig. 5e. This HCP lattice was generated via gliding partial dislocations by a single atomic layer away from SF3, and the continuous generation of SFs a single atomic layer away from the HCP phase boundary was repeated during straining, which caused the growth of the HCP phase, as shown in Fig. 5f. Our results provide direct evidence that the FCC-HCP phase transition occurs via partial dislocation emission on alternating close-packed (111) planes, i.e., SFs that are one atomic layer away from each other, which differs from the Bain strain path that proceeds via "principal axis" straining[75]. With further straining, the HCP phase transformed into an FCC lattice via the emission of partial dislocations between the two separated SFs, and the FCC lattice in this region was twin-related to the matrix, as shown in Fig. 5g, h. This deformation twin formation via the FCC–HCP–FCC transition is rarely observed in

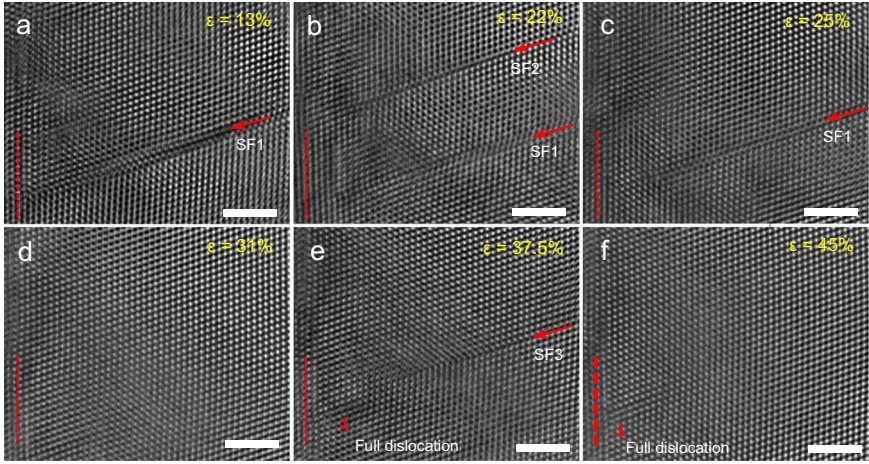

**Fig. 4 | In situ observation of partial dislocation emission and sliding of AuCu alloy NWs. a**, **b** the leading partial dislocations are emitted, which results in the SFs. **c**–**f** The trailing partial dislocation was generated on the same glide plane of the leading partial dislocations, which erased the pre-existing SF or formed full dislocation (marked with ⊥). The scale bar is 2 nm.

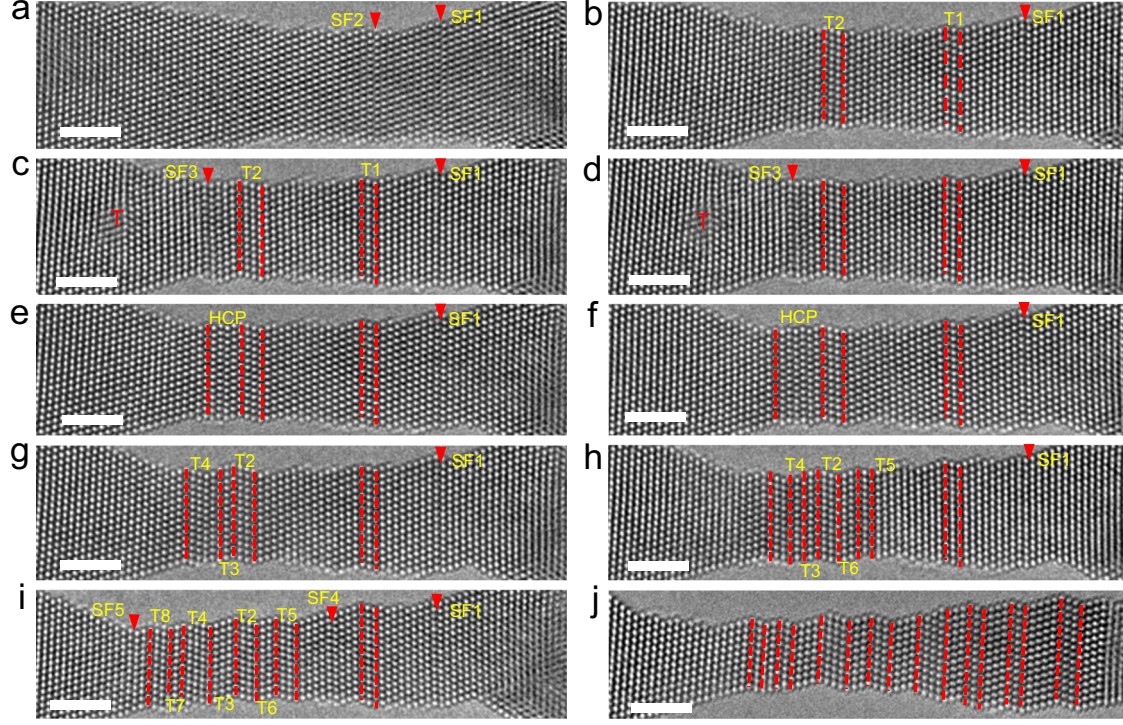

**Fig. 5 | In situ observation of the recoverable face-centered cubic (FCC)-hexagonal close-packed (HCP) phase transition and the formation of a zigzag-like nanotwin structure. a**–**f** FCC–HCP phase transition occurs via partial dislocation emission on alternating close-packed (111) planes. **g**, **h** HCP phase restores to an FCC lattice via the emission of partial dislocations between the two separated SFs. **i**, **j** Nanotwins were randomly generated throughout the NW, leading to a zigzag-like nanotwin structure. The stacking faults (SFs) are denoted SF1–SF3; the nanotwins are denoted T1–T6; and the twin boundaries are marked by the red dotted lines. The scale bar is 2 nm.

nanosized alloys. With extensive loading, as shown in Fig. 5i, j, nanotwins and FCC–HCP–FCC transitions were randomly generated throughout the NW, leading to the NW becoming a special zigzag-like morphology that contained a high density of coherent twin boundaries. These twin phase boundaries can impede dislocations and act as repulsive forces on dislocations, which leads to the strength of the NWs approaching ~6 GPa (Fig. 3d).

Figure 6 provides a sequence of enlarged HRTEM images showing the 2H phase generation and FCC–HCP–FCC phase transition process more clearly. From Fig. 6a, b, an SF (denoted SF1) that was four atomic layers away from the pre-existing nanotwin (denoted T1) was generated via partial dislocation emission. With loading, a new SF (denoted SF2), a single atomic layer away from SF1, led to a change in stacking sequence from ABCABCABCA to ABCABABABC, as shown in Fig. 6c. This change indicated that the FCC–HCP phase transition occurred, and the HCP phase exhibited a typical 2H structure with an AB stacking sequence. The generation of SFs a single atomic layer away from the HCP phase boundary was repeated with further straining, leading to growth of the HCP phase, as shown in Fig. 6d. As the strain increased, some of the HCP phase underwent a restoration process, leading to the

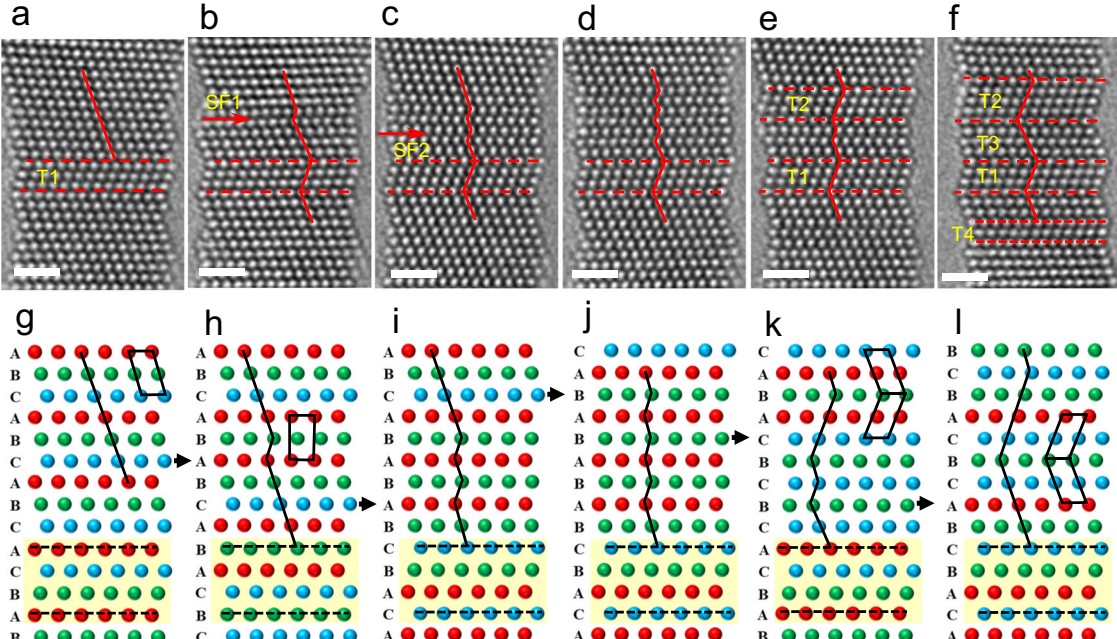

**Fig. 6 | In situ observation of recoverable FCC-2H phase transition. a–f** A sequence of enlarged HRTEM images showing the 2H phase generation and FCC–HCP–FCC phase transition process. **a–c** A series of enlarged HRTEM images shows that the generation of partial dislocations a single atomic layer away from each other leads to the HCP (2H) phase formation. **d** The generation of SFs a single atomic layer away from the HCP phase boundary was repeated with further straining, leading to the growth of the HCP phase. **e, f** This restoration of the FCC–HCP phase occurred via activation of the partial dislocations in the HCP phase. SF1 and SF2 represent stacking faults (SFs), and nanotwins are marked T1–T4. The scale bar is 1 nm. **g–l** Schematic illustrations of the process to show the recoverable FCC–HCP phase transition more clearly.

generation of a three-atomic-layer nanotwin in the NW (denoted T2), as shown in Fig. 6e. This restoration of the FCC–HCP phase occurred via activation of a partial dislocation in the HCP phase. In Fig. 6f, the residual HCP above T1 underwent a restoration process via partial dislocation emission and transformed into FCC lattices that were the same as the matrix of the NW. This partial dislocation with the Burgers vector should be in the direction opposite to that of the original dislocation. Therefore, an FCC–HCP-nanotwin transformation was realized during deformation. Figure 6g–l shows schematic illustrations to explain the FCC–HCP phase transition more clearly. As shown in Fig. 6g–j, partial dislocations with a single atomic layer away from each other cause the stacking sequence to change from ABCABCAB in an FCC lattice (Fig. 6g) to ABABABAB in a local HCP lattice (Fig. 6h–j). With further straining, partial dislocations with a Burgers vector opposite in direction to the original dislocation were generated, leading to the restoration of the FCC–HCP phase transition. Because the partial dislocations have different Burgers vectors from the original dislocations, the HCP–FCC transition also leads to twinning, which ultimately realizes an FCC matrix–nanotwin transformation.

In addition to the HCP phase with a 2H stacking sequence, the generation of the 4H phase with an ABCB stacking sequence was also observed. Figure 7 shows a sequence of enlarged HRTEM images of the 4H phase and multiple sub-nanosized twin formation processes. Figure 7a shows an HRTEM image captured after several nanotwin generations. Several nanotwins and two SFs (denoted SF1 and SF2) were generated in the NW. The nanotwins are denoted T1–T5. Because there is no obvious change during the subsequent deformation, T1–T5 serve as references. As the strain continuously increased, as shown in Fig. 7b, two new nanotwins were observed (labeled T6 and T7). In addition, a separated partial dislocation (SF3) of two atomic layers from the nanotwin T4 was also generated. In Fig. 7c, new partial emission on the adjacent plane of SF3, leading to local stacking sequence changes from ABCABCAB to ABCBABCB, exhibited a 4H phase structure, as marked by blue lines. With further straining, the generation of two partial

dislocations on the left side of the 4H phase boundary led to the formation of a four-atomic-layered twin (T8), and the 4H phase was restored to the FCC structure. Simultaneously, the twin boundary of T6 migrated an atomic layer to T7, forming a new 4H phase, as shown in Fig. 7d. With further straining, partial dislocation emission on the adjacent plane of twin boundaries (T8, T6) led to repeated FCC–4H phase transition and 4H phase boundary migration, as shown in Fig. 7e, f. This unconventional deformation model ultimately leads to the NW having a zigzag-like structure. In this deformation stage, the strength of the NW dramatically increased to ~6 GPa, indicating the strong strength hardening effect of these twin boundaries and HCP structures, including the 2H and 4H phases.

To reveal the underlying mechanism of plastic deformation, we employed MD simulations to study the shear deformation of the RSS Au$_{60}$Cu$_{40}$ alloy (see the "Methods" section and "Discussion" section of Supplementary Material for simulation details). Au and Cu atoms are randomly distributed in a supercell with dimensions of 9.5 nm × 5.5 nm × 26 nm ([11$\bar{2}$], [1$\bar{1}$0], and [$\bar{1}$11] directions, respectively). Under shear deformation, intrinsic SFs (ISFs) and extrinsic SFs (ESFs) are activated first by the gliding of a 1/6<112>-type Shockley partial dislocation on adjacent {111} planes. We observed the formation of very narrow deformation twins and HCP lamellae (including 2H and 4H) with a typical thickness of only a few nanometers, as shown in Fig. 8. Interestingly, the already formed nanoscale twin and HCP phases rarely broadened with increasing shear strain. Instead, new deformation twins and HCP phases emerged at other nucleation sites. The inset of Fig. 8 shows the detailed transition process of the nanoscale HCP lamellae to twins by shifting the stacking sequence, corresponding to the reversible FCC–HCP–FCC phase transition observed in Fig. 6. This deformation scenario is consistent with our in situ experimental observations, which is contrary to the continuous thickening of twins and phase transformations observed in pure metals. We also simulated the uniaxial tensile deformation in AuCu NW (Supplementary Fig. 11), and the results agree with the pure shear simulation results and our

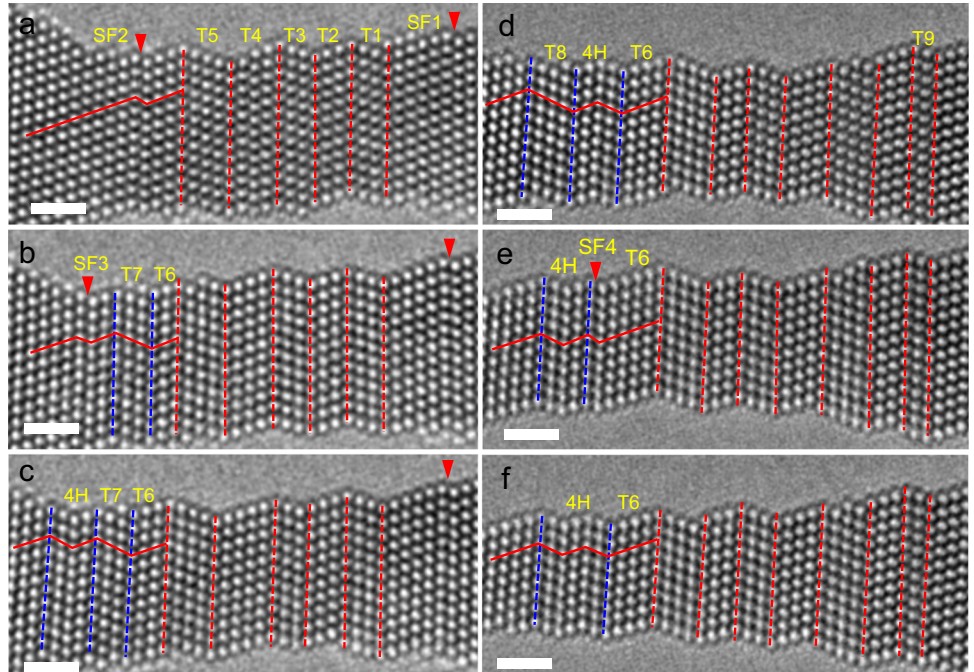

**Fig. 7 | In situ observation of FCC-4H phase transition. a–c** Enlarged HRTEM images showing the formation of FCC–4H phase transition. **d–f** partial dislocation emission on the adjacent plane of twin boundaries led to repeated FCC–4H phase transition and 4H phase boundary migration. The stacking faults (SFs) are marked SF1–SF4, and the nanotwins are marked T1–T10. The blue and red lines represent the boundaries. The scale bar is 1 nm.

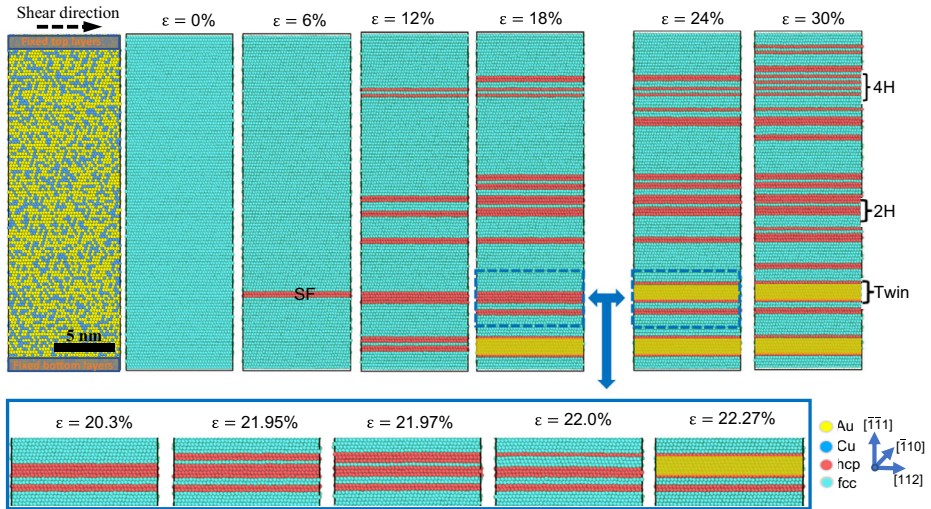

**Fig. 8 | Molecular dynamics (MD) simulation of shear deformation of random solid solution (RSS) AuCu alloy.** The formation of very narrow deformation twins and HCP lamellae (including 2H and 4H) with a typical thickness of only a few nanometers is observed. The atoms are colored according to common neighbor analysis: sky blue, red, and white atoms represent FCC, HCP, and other structures, respectively. Nanotwins are indicated by yellow shades, and stacking faults (SFs) and hexagonal close-packed (HCP) phases (including 2H and 4H) are also labeled. The scale bar is 5 nm. The inset indicates the detailed transition process of HCP to twin.

experimental observation. An example of MD-simulated pure Au under shear deformation is shown in Supplementary Fig. 12, where immediate thickening occurs after the nucleation of a twin. This indicates that the chemical inhomogeneity of the RSS atoms can significantly affect dislocation nucleation and motion in the AuCu alloy.

To unveil the intrinsic mechanism of dislocation nucleation for nanoscale twins in RSS AuCu, we employed the nudged-elastic-band (NEB) method to investigate the generalized SF energy (GSFE) surfaces, as illustrated in Fig. 9a, b (see "Methods" section for simulation

details). Figure 9a illustrates a schematic diagram of the formation of deformation twins from the initial FCC AuCu alloy by gliding the $1/6<112>$-type Shockley partial dislocations on adjacent {111} planes. Specifically, the first Shockley partial creates ISFs. The subsequent glides of the second and third Shockley partials on the neighboring {111} plane form an ESF and a three-layer twin region separated by two twin boundaries, respectively. The energy barriers for the three processes of FCC-to-ISF, ISF-to-ESF, and ESF-to-twin are denoted as $\Delta\gamma_1$, $\Delta\gamma_2$, and $\Delta\gamma_3$, respectively (see Fig. 9b, averaged over 250 independent

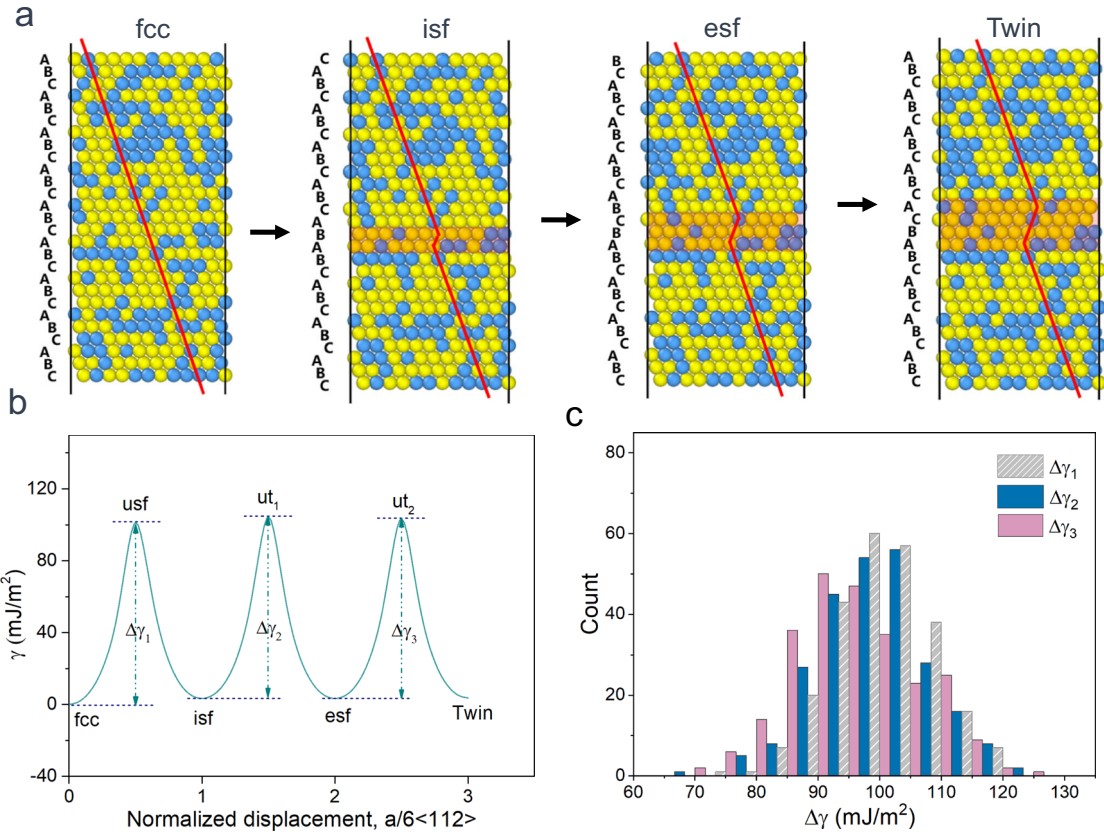

**Fig. 9 | The stacking fault energy calculation of AuCu alloy NW. a** Schematic diagram of the twin formation pathway in the AuCu alloy. Starting from the initial face-centered cubic (FCC) lattice, a one-layer intrinsic stacking fault (ISF), two-layer extrinsic stacking fault (ESF), and three-layer twins are created by introducing Shockley partials with $\mathbf{b} = 1/6<112>$ on the adjacent {111} plane. **b** Generalized stacking fault energy (GSFE) curve of the RSS AuCu alloy along the twin formation pathway. **c** The distribution of unstable planar fault energies for three selected paths: FCC-to-ISF, ISF-to-ESF, and ESF-to-twin, corresponding to $\Delta\gamma_1$, $\Delta\gamma_2$, and $\Delta\gamma_3$ in (b), respectively.

AuCu samples). For metals and conventional alloys, the following relationship usually exists $\Delta\gamma_1 > \Delta\gamma_2 = \Delta\gamma_3$. For pure Au here, $\Delta\gamma_1 = 68$ mJ/m², $\Delta\gamma_2 = \Delta\gamma_3 = 5$ mJ/m² [76]. In this case, it is always easier to thicken an existing twin than to nucleate a new SF in a pristine lattice, as shown in Supplementary Fig. 12. However, for the RSS AuCu alloy, in which solid-solution atoms are inhomogeneously distributed, the shearing of the neighboring {111} plane is subjected to a locally varying chemical environment, resulting in non-singular energy barriers. As shown in Fig. 9c, there are wide distributions of unstable planar fault energies for the three selected paths, $\Delta\gamma_1$, $\Delta\gamma_2$, and $\Delta\gamma_3$, among the 250 independent AuCu samples. The distributions of $\Delta\gamma_3$ (98.3 mJ/m²) and $\Delta\gamma_3$ (97.4 mJ/m²) are overall smaller than that of $\Delta\gamma_3$ (99.4 mJ/m²), and the former two distributions largely overlap. Such widely distributed and overlapping energy barriers enable the concurrent activation of separated partial dislocation, HCP phase transformation, and deformation twin generation. This is because the continuous thickening of an existing nanotwin would require a lower energy barrier for the successive shearing of neighboring {111} planes. The chance of this occurring would be low, considering the significantly varied energy barriers for twin thickening owing to the RSS-induced inherent chemical inhomogeneity. Once a plane with a high energy barrier is encountered, the thickening stops and the strain is carried by an SF (hence, a new twin) nucleated elsewhere at a new site with low $\Delta\gamma_1$. Thus, a high density of HCP nanolamellae and separated sub-nanoscale deformation twins are formed instead of individual thick twins in the RSS AuCu NW. This explains the experimental observations during our in situ nanoscale deformation and large-scale MD simulation.

Based on the in situ experiments and MD simulation, the inherent chemical inhomogeneity from an RSS alloy can lead to

energy barriers that overlap in magnitude for multiple plasticity mechanisms, thus rendering them equally competitive in RSS AuCu NWs. These mechanisms include separated SFs, HCP nanolamellae (2H and 4H phases), zigzag-like nanotwins, and ordinary dislocation activities on multiple slip systems, which is very different from the expected thickening of a single twin across the gauge section in a pure FCC metal. Previous MD simulations and experiments have shown that the elastic strain of metallic NWs can approach as high as ~7% [36,48,54,59], always along with relatively low ductility (also see dark green dots in Fig. 3e). Such mechanism results in an ultrahigh elastic strain of ~5.5% and superplasticity of ~260%, which has not been approached in Au [13,15,17,45–47] or Cu [16,23,25,60]. This distinctive behavior can possibly be understood by considering three factors. First, the concurrent activation of multiple plasticity mechanisms can effectively prevent the reorientation caused by the FCC matrix from directly transforming into nanotwins, eliminating necking localization and resulting in superplasticity. Second, the generated HCP nanolamellae and nanotwins (i.e., Stage 2) can serve as a proficient strengthening source for the pronounced strain hardening. Specifically, previous studies have predicted that twin and phase boundaries could exert a repulsive force on gliding dislocation, which can be described by $\tau \propto 1/\lambda$, where $\lambda$ is the thickness of the twin or HCP phase [77,78]. With ultrahigh density boundaries, dislocation slip is confined in such a way that the elastic strain energy cannot be easily released by means of dislocation propagation. Therefore, a higher stress is required for the nucleation and slip of dislocations for NWs with subaerial-scale HCP nanolamellae and subaerial-scale twins (as shown in Figs. 5–7). Third, the RSS-induced inherent chemical inhomogeneity leads to the forward migration of

the dislocation line, which undergoes a wavy and sluggish process during tensile deformation.

Previous studies on pure metals have shown that there is a size effect on the deformation mechanism: the deformation mechanism will switch from full or extended dislocation to partial dislocation resulting in deformation twinning as the size decreases. In contrast, the observed dual-stage plastic deformation mechanism in the current study should be the result of a combination of the size effect and inherent chemical inhomogeneity in AuCu. Based on previous theories[79,80], extended and full dislocations consist of leading followed by trailing partial dislocations, where the distance $r$ between the leading and trailing partial dislocations can be affected by the shear stress $\sigma$ according to the formula:

$$r = \frac{\kappa_1 \mathbf{b}^2 / \gamma}{1 - \sigma / \kappa_2 \gamma} \qquad (1)$$

where $\gamma$ is the SF energy of the metals, and $\kappa_1$, $\kappa_2$ are the constants depending on the elastic moduli of the material. At Stage 1, the size of the NW is relatively large, and the corresponding stress is relatively low (the value of $r$ is relatively small). It is reasonable that the plastic deformation was governed by leading followed by trailing partial dislocations forming extended dislocations or full dislocations (Fig. 3a, Supplementary Figs. 4 and 5). This deformation is similar to our previous observations in large-sized alloys[41] because the size of the NWs and the stress values are similar. In contrast, at Stage 2, the deformation mechanism is different from our previously reported observations[41] because both the size and stress have changed obviously. One can see that the size of the AuCu NW is ~5 nm, and the stress is above ~4.0 GPa. The increasing stress increases the value of $r$, leading to the leading partial dislocations quickly escaping to the surface before trailing partial dislocations nucleation; thus, there is no chance of forming a full dislocation or extended dislocation in such a small NW. The inherent chemical inhomogeneity also facilitates the concurrent activation of separated partial dislocation, HCP phase transformation, recurrence of reversible FCC–HCP phase transition, and zigzag-like nanotwin generation. Thus, the deformation mechanism of alloys should consider both the size effect and RSS-induced inherent chemical inhomogeneity, and this should be unique for RSS alloys. Such a deformation mechanism has rarely been reported in previous studies and may result from the following three factors. First, most previous studies were carried out on pure metals, in which the deformation was governed by layer-by-layer twin generation. Second, previous postmortem observations did not include the atomic-scale structure evolution process and thus could not detect the reversible process. Third, most previous studies were carried out on relatively large samples, thus the size effect and inherent chemical inhomogeneity-induced dual-stage plastic deformation mechanism could not be observed.

Before closing, we note that the binary AuCu in this work can be regarded as a prototype of highly concentrated solid-solution alloys, in which the random distribution of different elements inevitably renders inherent chemical inhomogeneity, engendering additional variables in which the atomic arrangement on each slip plane is different. This would naturally imply similar plastic deformation mechanisms for solid-solution alloys with multiple principal elements, that is, polycrystalline structured high/medium-entropy alloys, which have experienced a worldwide surge in research interest in recent years. Previous studies have observed that narrow twins, and possible HCP lamellae, typically only a few nanometers in thickness, are prevalent in several deformed high/medium-entropy alloys, such as in CrCoNi[81–85] and CrMnFeCoNi[86–88]. These studies suggest that deformation-induced nanotwins and HCP nanolamellae play an important role in the excellent fracture toughness of these alloys[81,86,89]. The current results, together with previous classic experiments and simulations[81–91], reveal

that the concurrent activation of multiple unconventional plasticity mechanisms could be generalized to highly concentrated solid solutions, from binary alloys to high/medium-entropy alloys. The observed deformation mechanism in single-crystal RSS AuCu NW is expected to operate in bulk high/medium-entropy alloys. However, it should be noted that most previous studies were carried out on polycrystalline alloys, which have a large number of grain boundaries. These grain boundaries can obstruct dislocation motion and contribute to strength. Thus, grain boundary-induced strengthening should be absent in the current study, whereas it is expected to operate in those polycrystalline bulk alloys.

In summary, the atomic-scale tensile processes of AuCu NWs were captured in situ using a custom-made device. Our results show that chemical inhomogeneity enables the highly concentrated solid-solution AuCu NWs to overcome the classic conflict between strength and ductility. The FCC–HCP phase transition, zigzag-like nanotwins, restoration of FCC–HCP phase transition, and ordinary dislocation activities on multiple slip systems all contribute to the superplasticity of the NWs. Plastic deformation generated HCP phase transitions and nanotwins that obstructed the dislocation motion, and the solid solution structure caused wavy and sluggish dislocation motion, leading to ultrahigh strength. Experimental and atomistic simulation results revealed that this behavior originates from inherent chemical inhomogeneity, in which the locally varying chemical environment enables the concurrent activation of multiple plasticity mechanisms. This helps shed light on the plastic deformation mechanism of these highly concentrated solid-solution alloys and is of practical value for the design of metallic materials with excellent mechanical properties.

## Methods

### In situ experimental procedure

The in situ experiment was conducted using a specially designed double-tilt deformation stage (Bestron-TEM in situ holder), which was operated in an FEI Titan environmental-TEM (E-TEM, 300 kV). An AuCu film with a thickness of ~30 nm was deposited on a single-crystal NaCl substrate via magnetron sputtering. The film with the substrate was attached to a bimetallic strip extensor using epoxy resin. The NaCl substrate was then etched away with deionized water, leaving a free-standing AuCu thin film on the bimetallic extensor. To prepare the AuCu NW, the AuCu film was first thinned using a precision ion polishing system, forming many variously sized nanovoids. The ligaments between neighboring nanovoids served as AuCu NWs. The NWs were then processed using a Fischione 1040 NanoMill to further thin them and remove the amorphous and carbon deposits around the NWs. Finally, atomic-scale in situ tensile experiments were conducted on the appropriate AuCu NWs using E-TEM.

### Calculation of 2D and 3D element mapping

The atomic-scale energy-dispersive X-ray spectroscopy (EDS) mapping of Cu/Au and the corresponding HAADF image were simultaneously captured at the same position. Therefore, the atomic column coordinates in the atomic-scale EDS mapping of Cu/Au were the same as those in the corresponding HAADF images. For HAADF images, the center position of each atom column was calculated using the open-source software Atomap. Then, the position coordinates of the atom columns were mapped to the EDS mappings, which were well-matched, as shown in Supplementary Fig. 13.

For the atomic-scale EDS mappings of Cu and Au, the average intensity of the image pixels was calculated within a small neighborhood of the atomic coordinates. This intensity value represents the relative intensity of the element in the atom column and the relative intensities of Au and Cu were $A_i$ and $C_i$, respectively. The relative content was calculated based on the relative intensities of Au and Cu.

The formula is as follows:

$$P_i = \frac{C_i}{A_i + C_i} \tag{2}$$

where $P_i$ represents the relative percentage content of Cu in the atom column.

Based on the $P_i$ and the atom column coordinates, we drew two-dimensional images of the Cu (and Au) percentage content, as shown in Fig. 1 and Supplementary Fig. 2. The different colors represent different content. Then, a two-dimensional Gaussian fitting was performed on each atom column to obtain a three-dimensional image (Fig. 1f). Greater height represents a higher content of Cu at the atom column.

## Shear deformation simulations

To reveal the underlying deformation mechanism, MD simulations were conducted on the RSS $Au_{60}Cu_{40}$ alloy using large-scale atomic/molecular massive parallel simulator (LAMMPS) packages[92]. The shear deformation of the $Au_{60}Cu_{40}$ alloy was simulated in a 9.5 nm × 5.5 nm × 26 nm orthogonal simulation cell containing 96,000 atoms with the $x$, $y$, and $z$ axes along the [112], [110], and [$\bar{1}11$] directions, respectively. Au and Cu atoms were randomly assigned to the crystal lattice of the supercell to model an RSS. The Au–Cu embedded-atom method potentials developed by Gola et al. were employed for our MD simulations[93]. Periodic boundary conditions were imposed along the $x$ and $y$ directions, and shrink-wrapped non-periodic boundary conditions were imposed in the $z$ direction. After relaxation at 300 K for 20 ps, the loading setup was arranged so that the bottom two layers of atoms (with a thickness of 0.8 nm) in the $z$ direction were fixed, and the top two layers of atoms were treated as a rigid body. Subsequently, shear deformations were performed by displacing the uppermost frozen atomic planes along the [112] direction at a shear velocity of 0.01 Å/ps. Deformation data were collected for the following 4 ns under the NVT ensemble with a time step of 2 fs[94].

## The GSFE calculations

A total of 250 independent AuCu samples with planar fault areas of 2 nm × 2 nm were used to calculate the GSFE energetics. Starting from the initial FCC lattice, a one-layer ISF, two-layer ESF, and three-layer twins were created by introducing Shockley partials with $\mathbf{b} = 1/6<112>$ on the adjacent {111} plane. The GSFE pathways were calculated using the NEB method performed in LAMMPS. Fifteen intermediate images were embedded uniformly into the corresponding initial and final states (i.e., FCC-to-ISF, ISF-to-ESF, and ESF-to-twin pathways). The NEB calculation was stopped when the change in energy of the minimization iterations was less than $1 \times 10^{-14}$ eV.

## Data availability

The data that support the findings of this study are available from the corresponding author upon request.

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

## Acknowledgements

This work was supported by the National Key R & D Program of China (2021YFA1200201); Beijing Outstanding Young Scientists Projects (BJJWZYJH01201910005018), the Natural Science Foundation of China (12174014) and the Basic Science Center Program for Multiphase Evolution in Hypergravity of the National Natural Science Foundation of China (51988101).

## Author contributions

X.D.H. and L.H.W. designed the project and guided the research. C.P.Y. conducted the in situ TEM experiments. C.P.Y., J.T, L.B.F., Z.X.W., R.W.S, Z.Q.W., and X.X.C analyzed the experimental results and wrote the initial draft. J.D. and B.Z.Z. performed MD simulation and related discussion. X.D.H. and L.H.W. finalized the paper. All authors contributed to extensive discussions of the results.

## Competing interests

The authors declare no competing interests.
