## [Peer Review File · Nature Communications]

Chemical inhomogeneity–induced profuse nanotwinning and phase transformation in AuCu nanowiresREVIEWER COMMENTS

Reviewer #1 (Remarks to the Author):

In the work, the authors performed in situ tensile straining and MD simulations to study the deformation behavior of Cu-Au alloy at the nanoscale. The authors observed superplasticity of ~260% and profuse HCP phase generation. The work is nicely done and the manuscript very well written. I recommend accepting the paper if the authors can successfully address my concerns.

1. Lack of baseline. I recommend the authors to perform the same experiments on pure Cu or Au and plot the curves together to show that the Cu-Au indeed has improved ductility.
2. Effect of e-beam. We do not know whether the exceptional ductility is caused by the electron beam irradiation or it is something intrinsic in the material. I recommend the authors turn off the beam and perform the tensile test again in TEM to exclude the effect of e-beam.

Reviewer #2 (Remarks to the Author):

Nanosized metals have attracted extensive interest because of their high strength. Because the mechanical properties are directly related to the atomic-scale deformation mechanism of nanosized metals, large number of studies have been carried out in the past decades. Previous studies show that nanosized metals always exhibit a strength-ductility trade-off phenomenon. However, most of the previous studies were carried out on pure metals. For highly concentrated solid-solution alloys, the in situ atomic-scale experimental evidence is rarely approached. No doubt that the topic of in situ atomic-scale deformation process of nanosized alloy is appealing. In this manuscript, the authors show the FCC AuCu alloy NWs exhibit superplasticity of ~260% and ultra-high strength of ~6 GPa, overcoming the trade-off between the strength and ductility. They provide the in situ atomic-scale evidence that the HCP phase generation, reversible FCC-HCP phase transition and zigzag-like nanotwins generation are all contribute the observed superplasticity and ultra-high strength. They suggested that the observed deformation modes were original from the randomly distributed solid solution atoms induced chemical inhomogeneity. The results are interesting and the present observations are certainly of high quality. However, there are some comments must be addressed before this interesting paper publication.

1. At current study, the NWs is AuCu alloy, while from the Fig. 1, the author only provide the element mapping of Cu. The reviewer suggested the author also provide the element mapping of Au, this is important for the readers understanding the chemical inhomogeneity of such alloy.
2. The reviewer suggested the author also provide more details of the 3D element mapping, and how this date was obtained from the mapping.
3. From the HAAD image, the region with high ratio of Au should be brighter than those regions with high ratio of Cu, the authors should provide direct evidence of this.
4. From the Fig. 2a, 2b, there are high density of partial dislocation resulted SFs in the NW, while in Fig.

2c, the NW seems is defect free. How these SFs disappeared is very important because this is directly related to the deformation mode at this early stage.

5. From the Fig. 3d, the measured elastic strain is more than 5.5%, this large value seems rarely approached, the author should add a short discussion compare with previous studies, and provide more details explanation.

6. From the Fig. 7a, the T1-T3 seems constructed a special structure, the reviewer suggested that authors re-check this structure.

7. For the MD simulation, it's well established that the deformation mode can significantly affect by the potentials that used. How the authors make sure the potentials that used is valid for AuCu should be verified. How this sample constructure by MD simulation should be provided.

8. The author claims that the deformation modes in AuCu can explain the exceptional fracture toughness in high/medium-entropy alloys, are there any similar deformation modes also observed in high/medium-entropy alloys?

9. The author claims that the deformation modes in AuCu can explain the exceptional fracture toughness in high/medium-entropy alloys, while it should be noted that most of previous studies were carried out on polycrystalline with GBs, the strengthening mechanism should be different, the author should make a clear statement.

10. The observed reversible FCC-HCP phase transition and zigzag-like nanotwins is very interesting, why previous studies have not observed this phenomenon? The reviewer suggested the author provide a short discussion and highlight the advances of their discovery compare with previous reports. This related discussion will make the paper more comprehensive and more attractive.

Reviewer #3 (Remarks to the Author):

This manuscript presents a very nice work on the in situ investigation of the atomic-scale plastic deformation mechanisms of AuCu nanowires. The in situ TEM work is of high quality, clearly illustrating that the remarkable mechanical performance of AuCu nanowires arising from the activation of multiple plasticity mechanisms, including dislocation activities on multiple slip systems, stacking faults, reversible FCC-HCP-FCC phase transformations and nanotwins. The authors further conducted MD simulations, revealing that the local fluctuation of energy barriers for different deformation mechanisms caused by chemical inhomogeneity is the intrinsic reason for the concurrent activation of these multiple plasticity mechanisms. I regard this study a very valuable contribution to the field of RSS alloys. In summary, the work is convincing and of interest to the community. However, there are several concerns regarding about data analysis and the interpretations of the mechanisms that should be addressed to justify its publication in Nature Communications.

1. There is a question related to the calculation of the stress value in each tensile deformation. The stress value is estimated by the elastic strain along the tensile direction ($[111]$), while the elastic strain is calculated by measuring the distance of (111) plane diffraction spots in the corresponding FFT pattern.

To my knowledge, there is a significant error (greater than 1%) in measuring the interplanar spacing from FFT patterns, because the diffraction spots in FFT usually do not have regular shapes. The authors determine the center of the diffraction spots by plotting the intensity profile of the straight lines passing through two symmetrical diffraction spots. Nevertheless, the width or position of the selected lines (such as the dashed rectangles in FFT patterns in Fig. S5) may significantly influence the measurement. The error bars (I guess they represent the errors for the calculated stress) shown in Fig. 3d are somehow smaller than expected. Please make it clear how the error value in each spot was determined? This could be important because the high stress (~ 6 GPa) is a main conclusion of this work.

2. The team members already published a related article about AuCu NW, "In situ atomic-scale observation of AuCu alloy nanowire with superplasticity and high strength at room temperature." This article was cited in the manuscript. However, there should be more discussions since they are strongly related. What are the similarities and differences in their behaviors? One is dislocation-dominated and the other is partial dislocation (SF and twinning) dominated? If so, what causes this difference? The authors should clarify on this matter.

3. The author claims the dual-stage plastic deformation, which contributes to the observed superplasticity and high strength. What is the intrinsic mechanism for the transformation between the dual-stage, and whether is this unique for RSS alloys?

4. For FCC lattice, stacking faults normally appear as two-layer HCP atoms. If there are several SFs stacking together, how can they be effectively distinguished from the FCC-HCP phase transition? Are there any possibilities that the phase transition mentioned in the article is just a pile-up of SFs?

5. MD simulations were applied to clarify the difference in between the deformation mechanisms of the RSS alloys and pure Au. However, it is not exact the same process as experiment (pure shear vs. uniaxial stress). It might be more convincing if the author can mimic similar loading process (uniaxial stress) in MD simulations.

Reply to the Reviewers

We thank the reviewers for their in-depth review. The constructive suggestions provided have been considered to substantially improve the paper. A point-by-point response is included below, in which the reviewer comments (in italic font) are first repeated and then our responses follow. The changes made in the revised manuscript are indicated by black bold font.

Reviewer #1:

General comments: In the work, the authors performed in situ tensile straining and MD simulations to study the deformation behavior of Cu-Au alloy at the nanoscale. The authors observed superplasticity of ~260% and profuse HCP phase generation. The work is nicely done and the manuscript very well written. I recommend accepting the paper if the authors can successfully address my concerns.

Response: We are grateful for the reviewer's immensely positive comments.

Comment 1: Lack of baseline. I recommend the authors to perform the same experiments on pure Cu or Au and plot the curves together to show that the Cu-Au indeed has improved ductility.

Response: Following the reviewer's constructive suggestion, we have performed the tensile experiments on Au nanowire. The experimental results and corresponding curve were added in Fig. S12 because we noticed that adding the curve of Au into the current Fig. 3 made it very difficult to distinguish (see **Fig. R1** below). In addition, a short discussion was added in the revised manuscript as follows:

“In addition, we conducted tensile experiments on Au NWs (as indicated in Fig. 3e), which exhibited the homogeneous plasticity of ~85% and strength of ~2.3 GPa (also see Fig. S12). These results are similar with previously reported experimental studies on Au [45]. Comparing Fig. 2 with Fig. S12, one can see that the RSS AuCu alloy NWs indeed overcome the strength–ductility trade-off”.

Fig. R1 Statistical data of the elastic strain as a function of the uniform elongation of Au and AuCu nanowires.

Fig. S12 Statistical data of the elastic strain as a function of the uniform elongation of Au nanowires. The scale bar is 5 nm.

Comment 2. *Effect of e-beam.* We do not know whether the exceptional ductility is caused by the electron beam irradiation or it is something intrinsic in the material. I recommend the authors turn off the beam and perform the tensile test again in TEM to exclude the effect of e-beam.

Response: Thanks for the reviewer’s constructive suggestion. We have performed the tensile experiments on AuCu nanowire under beam blank (only turning on the electron beam for image capture), and the images were captured at very low electron beam dose in order to reduce the influence of electron beam irradiation. The superelongation was also observed under these conditions in AuCu nanowire, as shown in Fig. S7.

In addition, several sentences were added in the revised manuscript, as follows:

“We also performed the tensile experiments of AuCu NWs under beam blank (only turning on the electron beam for image capture), while the super-elongation was still observed (as shown in Fig. S7), indicating that the electron beam irradiation had no significant impact on the exceptional ductility of AuCu NWs.”.

Fig. S7 Examples showing the super-elongation of random solid solution AuCu nanowires under tensile loading with beam blank.

Reviewer #2:

General comments: Nanosized metals have attracted extensive interest because of their high strength. Because the mechanical properties are directly related to the atomic-scale deformation mechanism of nanosized metals, large number of studies have been carried out in the past decades. Previous studies show

that nanosized metals always exhibit a strength-ductility trade-off phenomenon. However, most of the previous studies were carried out on pure metals. For highly concentrated solid-solution alloys, the in situ atomic-scale experimental evidence is rarely approached. No doubt that the topic of in situ atomic-scale deformation process of nanosized alloy is appealing. In this manuscript, the authors show the FCC AuCu alloy NWs exhibit superplasticity of ~260% and ultra-high strength of ~6 GPa, overcoming the trade-off between the strength and ductility. They provide the in situ atomic-scale evidence that the HCP phase generation, reversible FCC-HCP phase transition and zigzag-like nanotwins generation are all contribute the observed superplasticity and ultra-high strength. They suggested that the observed deformation modes were original from the randomly distributed solid solution atoms induced chemical inhomogeneity. The results are interesting and the present observations are certainly of high quality. However, there are some comments must be addressed before this interesting paper publication.

Response: We are grateful for the reviewer’s immensely positive comments.

Comment 1. At current study, the NWs is AuCu alloy, while from the Fig. 1, the author only provide the element mapping of Cu. The reviewer suggested the author also provide the element mapping of Au, this is important for the readers understanding the chemical inhomogeneity of such alloy.

Response: Thanks for the reviewer’s constructive suggestion. In response, we have provided the element mapping of Au in the Supplementary Materials (Fig. S2) and revised the manuscript accordingly.

Fig. S2 a, b Two- and three-dimensional element mapping of Au calculated from Fig. 1d and 1e demonstrating that the solid-solution atoms are randomly distributed in the AuCu.

Comment 2. The reviewer suggested the author also provide more details of the 3D element mapping, and how this data was obtained from the mapping.

Response: Thanks for the reviewer’s constructive suggestion. We have provided the detailed process of how the three-dimensional element mapping was obtained from the atomic-scale energy-dispersive X-ray spectroscopy (EDS) mapping in Methods section of the revised manuscript. The details are as follows:

“Calculation of 2D, 3D element mapping:

The atomic-scale energy-dispersive X-ray spectroscopy (EDS) mapping of Cu/Au and the corresponding high-angle annular dark-field (HAADF) image were simultaneously captured at the same position. Therefore, the atomic column coordinates in the atomic-scale EDS mapping of Cu/Au were the same as those in the corresponding HAADF images. For HAADF images, the center position of each atom column was calculated using the open-source software Atomap. Then, the position coordinates of the atom columns were mapped to the EDS mappings, which were well matched, as shown in Fig. S15.

For the atomic-scale EDS mappings of Cu and Au, the average intensity of the image pixels was calculated within a small neighborhood of the atomic coordinates. This intensity value represents the relative intensity of the element in the atom column, and the relative intensities of Au and Cu were A_i

and C_i , respectively. The relative content was calculated based on the relative intensities of Au and Cu. The formula is as follows:

$$P_i = \frac{C_i}{A_i + C_i}$$

where P_i represents the relative percentage content of Cu in the atom column.

Based on the P_i and the atom column coordinates, we drew two-dimensional images of the Cu (and Au) percentage content, as shown in Figs. 1 and S2. The different colors represent different content. Then, a two-dimensional Gaussian fitting was performed on each atom column to obtain a three-dimensional image (Fig. 1f). Greater height represents higher content of Cu at the atom column.”.

Fig. S15 Process of obtaining the position coordinates of the atom columns by the atomic-scale energy-dispersive X-ray spectroscopy (EDS) mappings and corresponding high-angle annular dark-field (HAADF) image.

Comment 3. From the HAADF image, the region with high ratio of Au should be brighter than those regions with high ratio of Cu, the authors should provide direct evidence of this.

Response: Thanks for the reviewer's suggestion. We have provided the HAADF image of AuCu alloy and corresponding 2D intensity mapping, as shown in Fig. S1. It can be seen from Fig. S1 that the intensity is diverse at different atomic column, which provides the direct evidence that those region with high ratio of Au exhibiting relatively high intensity.

Fig. S1 a High-angle annular dark-field (HAADF) image of AuCu alloy. b HAADF intensity mapping calculated from a, showing that there is different intensity at different atomic columns.

Comment 4. From the Fig. 2a, 2b, there are high density of partial dislocation resulted SFs in the NW, while in Fig. 2c, the NW seems is defect free. How these SFs disappeared is very important because this is directly related to the deformation mode at this early stage.

Response: In response to the reviewer’s constructive comments, several sentences were added in the revised manuscript, as follows:

“In Stage 1, the plastic deformation was mediated by extended dislocations and full dislocations that resulted from a leading partial dislocation followed by a trailing partial dislocation (see examples

in Figs. S4 and S5). These dislocations quickly escaped from the small NW with no debris, resulting in the nearly defect-free AuCu NW during loading (as shown in Figs. 2c, 2d, and 4).”.

Comment 5. From the Fig. 3d, the measured elastic strain is more than 5.5%, this large value seems rarely approached, the author should add a short discussion compare with previous studies, and provide more details explanation.

Response: Thanks for the reviewer’s suggestion. In response, a short discussion was added in the revised manuscript, as follows:

“Previous MD simulations and experiments have shown that the elastic strain of metallic NWs can approach as high as ~7% [36, 48, 54, 59], always along with relatively low ductility (also see dark green dots in Fig. 3e). Such novel mechanism results in an ultrahigh elastic strain of ~5.5% and superplasticity of ~260%, which has not been approached in Au [13, 15, 17, 45–47] or Cu [16, 23, 25, 60]. This distinctive behavior can possibly be understood by considering three factors. First, the concurrent activation of multiple plasticity mechanisms can effectively prevent the reorientation caused by the FCC matrix from directly transforming into nanotwins, eliminating necking localization and resulting in superplasticity. Second, the generated HCP nanolamellae and nanotwins (i.e., Stage 2) can serve as a proficient strengthening source for the pronounced strain hardening. Specifically, previous studies have predicted that twin and phase boundaries could exert a repulsive force on gliding dislocation, which can be described by $\tau \propto 1/\lambda$, where λ is the thickness of the twin or HCP phase [77, 78]. With ultrahigh density boundaries, dislocation slip is confined in such a way that the elastic strain energy cannot be easily released by means of dislocation propagation. Therefore, a higher stress is required for the nucleation and slip of dislocations for NWs with subaerial-scale HCP nanolamellae and subaerial-scale twins (as shown in Figs. 5–7). Third, the RSS-induced inherent chemical inhomogeneity leads to the forward migration of the dislocation line, which undergoes a wavy and sluggish process during tensile deformation.”.

Comment 6. From the Fig. 7a, the T1-T3 seems constructed a special structure, the reviewer suggested that authors re-check this structure.

Response: Thanks for the reviewer’s suggestion. We found that the nanotwin T1-T3 constructed a special 6H structure after re-checking Fig. 7a. However, they also can serve as three-atomic-layer twins; therefore, to avoid confusion with the discussion of FCC-4H phase transition, we didn’t make any related changes in the manuscript.

Comment 7. For the MD simulation, it’s well established that the deformation mode can significantly affect by the potentials that used. How the authors make sure the potentials that used is valid for AuCu should be verified. How this sample constructure by MD simulation should be provided.

Response: Following the reviewer’s suggestion, the detailed process of molecular dynamics simulation was added in the Supplementary Materials (see Discussion section), as follows:

“Discussion

Verification of the reliability of the potential used in this work

The reliability of the potential by Gola et al. used in the present work was verified in several ways. The enthalpies of mixing, lattice constants, planar energetics, etc. from this potential have been compared in detail with experimental and density functional theory (DFT) calculations (see Ref. 93 in the main manuscript) to demonstrate its satisfying accuracy. The stacking-fault energy (SFE) is one of the critical parameters that determines the deformation mode and mechanical behavior of alloys. We compared the calculated SFE of Au₆₀Cu₄₀ alloy as well as pure Au and Cu, using this potential (Gola et al.), DFT calculation and another empirical potential (Zhou et al., see Ref. 1 in the Supplementary

Materials) in Fig. S16. That showed the relative agreement of the average SFE of Au₆₀Cu₄₀ alloy between MD simulation (using Gola’s potential) and DFT calculation. We also employed Zhou’s potential to simulate the shear deformation of Au₆₀Cu₄₀ alloys (using the same setup as that in the main text), as shown in Fig. S17. We found the same nanometer-thick deformation twins and HCP phase, accompanied by some intrinsic and extrinsic stacking faults, compared to Fig. 8. Those results can demonstrate that the novel deformation mode revealed in present work is the intrinsic alloy property rather than the dependence of empirical potential for MD simulation.

Sample construction

To construct samples for shear simulation, we adopted a 9.5 nm × 5.5 nm × 26 nm orthogonal simulation cell containing 96,000 atoms, where the x, y, and z axes are along the [112], $[\bar{1}10]$, and $[\bar{1}\bar{1}1]$ directions, respectively. Periodic boundary conditions were applied to the x and y directions. The potentials selected in the current work were the embedded-atom method (EAM) potentials developed by Gola et al. (Ref. 1). We first froze the upper and lower atomic planes with a thickness of 0.8 nm and ran a total of 20 ps at 300 K under the NVT ensemble to make sure that the samples were fully equilibrated. Then, shear deformations were performed by displacing the uppermost frozen atomic planes along the [112] direction at a shear velocity of 0.01 Å/ps. The deformation data were collected for the following 8 ns under the NVT ensemble with time step of 2 fs.”

Fig. S16 Comparison of stacking faults energies calculated by density functional theory (DFT) and molecular dynamics (MD) simulation (using Gola’s potential and Zhou’s potential), respectively.

Fig. S17 Nanostructure evolution of bulk sample during successive shear deformation by molecular dynamics simulation

using Zhou's potential (Ref. 1), of which three different strain states are shown.

Comment 8. The author claims that the deformation modes in AuCu can explain the exceptional fracture toughness in high/medium-entropy alloys, are there any similar deformation modes also observed in high/medium-entropy alloys?

Response: Thanks for the reviewer's comment. According to the previous studies of high/medium-entropy alloys, similar deformation modes have been observed in high/medium-entropy alloys. In response, a short discussion has been added to the manuscript.

“Previous studies have observed that narrow twins, and possible HCP lamellae, typically only a few nanometers in thickness, are prevalent in several deformed high/medium-entropy alloys, such as in CrCoNi [81–85] and CrMnFeCoNi [86–88]. These studies suggest that deformation-induced nanotwins and HCP nanolamellae play an important role in the exceptional fracture toughness of these alloys [81, 86, 89].”.

Comment 9. The author claims that the deformation modes in AuCu can explain the exceptional fracture toughness in high/medium-entropy alloys, while it should be noted that most of previous studies were carried out on polycrystalline with GBs, the strengthening mechanism should be different, the author should make a clear statement.

Response: Thanks for the reviewer's constructive suggestion. In response, a short discussion was added in the revised manuscript, as follows:

“The observed deformation mechanism in single-crystal RSS AuCu NW is expected to operate in bulk high/medium-entropy alloys. However, it should be noted that most previous studies were carried out on polycrystalline alloys, which have a large number of grain boundaries. These grain boundaries can obstruct dislocation motion and contribute to strength. Thus, grain boundary–induced strengthening should be absent in the current study, whereas it is expected to operate in those polycrystalline bulk alloys.”.

Comment 10. The observed reversible FCC-HCP phase transition and zigzag-like nanotwins is very interesting, why previous studies have not observed this phenomenon? The reviewer suggested the author provide a short discussion and highlight the advances of their discovery compare with previous reports. This related discussion will make the paper more comprehensive and more attractive.

Response: Thanks for the reviewer's constructive suggestion. In response, a short discussion was added in the revised manuscript, as follows:

“Thus, the deformation mechanism of alloys should consider both the size effect and RSS-induced inherent chemical inhomogeneity, and this should be unique for RSS alloys. Such a deformation mechanism has rarely been reported in previous studies and may result from the following three factors. First, most previous studies were carried out on pure metals, in which the deformation was governed by layer-by-layer twin generation. Second, previous postmortem observations did not include the atomic-scale structure evolution process, and thus could not detect the reversible process. Third, most previous studies were carried out on relatively large samples, thus the size effect and inherent chemical inhomogeneity–induced dual-stage plastic deformation mechanism could not be observed.”

Reviewer #3:

General comments: This manuscript presents a very nice work on the in situ investigation of the atomic-scale plastic deformation mechanisms of AuCu nanowires. The in situ TEM work is of high quality, clearly illustrating that the remarkable mechanical performance of AuCu nanowires arising from the activation of multiple plasticity mechanisms, including dislocation activities on multiple slip systems, stacking faults,

reversible FCC-HCP-FCC phase transformations and nanotwins. The authors further conducted MD simulations, revealing that the local fluctuation of energy barriers for different deformation mechanisms caused by chemical inhomogeneity is the intrinsic reason for the concurrent activation of these multiple plasticity mechanisms. I regard this study a very valuable contribution to the field of RSS alloys. In summary, the work is convincing and of interest to the community. However, there are several concerns regarding about data analysis and the interpretations of the mechanisms that should be addressed to justify its publication in *Nature Communications*.

Response: We are grateful for the reviewer's immensely positive comments.

Comment 1. *There is a question related to the calculation of the stress value in each tensile deformation. The stress value is estimated by the elastic strain along the tensile direction ([111]), while the elastic strain is calculated by measuring the distance of (111) plane diffraction spots in the corresponding FFT pattern. To my knowledge, there is a significant error (greater than 1%) in measuring the interplanar spacing from FFT patterns, because the diffraction spots in FFT usually do not have regular shapes. The authors determine the center of the diffraction spots by plotting the intensity profile of the straight lines passing through two symmetrical diffraction spots. Nevertheless, the width or position of the selected lines (such as the dashed rectangles in FFT patterns in Fig. S5) may significantly influence the measurement. The error bars (I guess they represent the errors for the calculated stress) shown in Fig. 3d are somehow smaller than expected. Please make it clear how the error value in each spot was determined? This could be important because the high stress (~6 GPa) is a main conclusion of this work.*

Response: The reviewer is correct that the error of calculated elastic strain values is usually up to ~ 0.7% via measuring the interplanar spacing from FFT patterns (as shown in Fig. S8d). Before this paper submission, we had noticed this issue. In order to avoid this significant error, we have selected five different positions to measure the average elastic strain (as shown in Fig. S8d). With this method, the standard deviation shown in Fig. 3 is below 0.3% according the standard deviation formula $s = \sqrt{\frac{\sum_{i=1}^n (d_i - \bar{d})^2}{n-1}}$ calculated by averaging across multiple regions (see more details in Fig. S8d). This indicates that the error will be decreased sharply after multiple measurements.

To further verify the accuracy of the elastic strain values, we calculated the elastic strain by measuring the spacing of 15 lattices from HRTEM images, as shown in Fig. S9. The elastic strain could be calculated using $\varepsilon = (d_n - d_0)/d_0$, where d_0 is the lattice spacing before deformation, and d_n corresponds to the lattice spacing in the regions $n = 1, 2, 3, \dots, n$, as shown in Fig. S9d-h. These regions had no plastic deformation and were defect-free for a long time during the loading. The average elastic strain was ~3.6%, which is similar to the results calculated from FFT patterns (3.51%). In addition, we also calculated the elastic strain using IFFT images (Fig. S10), which gave a result (~3.63%) similar to the above calculated results. Therefore, our results are reliable within the error range.

We have provided the details of the multiple measurements of elastic strain in the Supplementary Material (Figs. S8-S10) and added several sentences in the revised manuscript, as follows:

“To evaluate the elastic strain more accurately, different methods were used to measure the elastic strain; the results show that the standard deviation is below 0.3%, and similar elastic strain values were obtained using different methods (as shown in Figs. S9 and S10). This indicates that the elastic strain values are relatively accurate and reliable.”

Fig. S8 Schematic illustrations showing the calculated process of elastic strain using the fast Fourier transform (FFT) of transmission electron microscopy images in AuCu nanowires (NWs). The lattice distance d_n of the high-resolution TEM (HRTEM) images can be calculated using the distance D_n between two symmetrical brightest diffraction spots in the FFT map. The lattice distance d_n can be easily obtained using $d_n = \frac{2}{D_n}$. Here, we only calculated the elastic strain along the tensile direction ([111]) by measuring the distance of (111) plane diffraction spots. Then, the elastic strain can be calculated by comparing the changes in distance between the strained data d_n and reference data d_0 according to the following formula: $\varepsilon_n(\%) = 100\% \times \frac{(d_n - d_0)}{d_0}$. a HRTEM image showing that the NW was fractured. The reference region was near the fractured region. The two symmetrical brightest diffraction spots are marked by two red arrows. b-d Calculated elastic strain under different tensile deformation of NWs.

Fig. S9 Elastic strain calculation by measuring the lattice distance of the NWs under loading at the different detect-tree regions. The elastic strain was calculated using $\varepsilon = (d_n - d_0)/d_0$. The average elastic strain was calculated as 3.6%, similar to the calculated results by fast Fourier transform. The error bar (standard deviation s) is 0.23%.

Fig. S10 Elastic strain calculation by inverse fast Fourier transform (IFFT) image of the (111) plane. a1–a3 Reference lattice distance measured using the IFFT image of the (111) plane. b1–b3 An example of elastic strain calculation using the IFFT image of the (111) plane.

Comment 2. *The team members already published a related article about AuCu NW, “In situ atomic-scale observation of AuCu alloy nanowire with superplasticity and high strength at room temperature.” This article was cited in the manuscript. However, there should be more discussions since they are strongly related. What are the similarities and differences in their behaviors? One is dislocation-dominated and the other is partial dislocation (SF and twinning) dominated? If so, what causes this difference? The authors should clarify on this matter.*

Response: Thanks for the reviewer’s construction suggestion. We have added a discussion in the revised manuscript, as follows:

“**At Stage 1, the size of the NW is relatively large, and the corresponding stress is relatively low (the value of r is relatively small). It is reasonable that the plastic deformation was governed by leading followed by trailing partial dislocations forming extended dislocations or full dislocations (Figs. 3a, S4, and S5). This deformation is similar with our previous observations in large-sized alloys [41] because the size of the NWs and the stress values are similar (as shown in Fig. S6 and S11). In contrast, at Stage 2, the deformation mechanism is different from our previously reported observations [41], because both the size and stress have changed obviously. One can see that the size of the AuCu NW is ~ 5 nm, and the stress is above ~ 4.0 GPa. The increasing stress increases the value of r , leading to the leading partial dislocations quickly escaping to the surface before trailing partial dislocations nucleation; thus, there is no chance to form a full dislocation or extended dislocation in such a small NW. The inherent chemical inhomogeneity also facilitates the concurrent activation of separated partial dislocation, HCP phase transformation, recurrence of reversible FCC-HCP phase transition, and zigzag-like nanotwin generation. Thus, the deformation mechanism of alloys should consider both the size effect and RSS-induced inherent chemical inhomogeneity, and this should be unique for RSS alloys.**”

Comment 3. *The author claims the dual-stage plastic deformation, which contributes to the observed superplasticity and high strength. What is the intrinsic mechanism for the transformation between the dual-stage, and whether is this unique for RSS alloys?*

Response: Thanks for the reviewer’s construction comment. At the current stage, we believe that the dual-stage plastic deformation mechanism should be the result of a combination of the size effect and inherent chemical inhomogeneity of RSS alloys, which should be unique for RSS alloys. For those pure metals, it is well established that there is a size effect on the deformation mechanism: the deformation mechanism will switch from full or extended dislocation (in large-sized metals) to partial dislocation resulting in deformation twinning as the size decreases. Our current observations of RSS AuCu NWs also show this trend. At stage 1,

the NW's size is relatively large, and the plastic deformation was governed by leading followed by trailing partial dislocations (as shown in Fig. 3a). These leading and trailing partial dislocations formed extended dislocations or full dislocations (also see Figs. S4 and S5), which is similar to previous observation in large sized alloys [41]. In contrast, at stage 2, the deformation mechanism is different from previously reported observations [41], because both the size and stress have changed obviously. One can see that the size of the NW is below ~5 nm, and thus the partial dislocations can quickly escape to the surface; thus, there is no chance nucleation of trailing partial dislocations to form full dislocation or extended dislocation. In addition, the inherent chemical inhomogeneity also facilitates the concurrent activation of separated partial dislocation, HCP phase transformation, recurrence of reversible FCC-HCP phase transition, and zigzag-like nanotwin generation.

Follow the reviewer's construction comments, we have added a discussion in the revised manuscript, as follows:

“Previous studies on pure metals have shown that there is a size effect on the deformation mechanism: the deformation mechanism will switch from full or extended dislocation to partial dislocation resulting in deformation twinning as the size decreases. In contrast, the observed dual-stage plastic deformation mechanism in the current study should be the result of a combination of the size effect and inherent chemical inhomogeneity in AuCu. Based on previous theories [79, 80], extended and full dislocations consist of leading followed by trailing partial dislocations, where the distance r between the leading and trailing partial dislocations can be affected by the shear stress σ according to the formula $r = \frac{\kappa_1 b^2 / \gamma}{1 - \sigma / \kappa_2 \gamma}$. At Stage 1, the size of the NW is relatively large, and the corresponding stress is relatively low (the value of r is relatively small). It is reasonable that the plastic deformation was governed by leading followed by trailing partial dislocations forming extended dislocations or full dislocations (Fig. 3a, S4, and S5). This deformation is similar with previous observations in large-sized alloys [41] because the size of the NWs and the stress values are similar (as shown in Fig. S6 and S11). In contrast, at Stage 2, the deformation mechanism is different from previously reported observations [41], because both the size and stress have changed obviously. One can see that the size of the AuCu NW is ~5 nm, and the stress is above ~4.0 GPa. The increasing stress increases the value of r , leading to the leading partial dislocations quickly escaping to the surface before trailing partial dislocations nucleation; thus, there is no chance to form a full dislocation or extended dislocation in such a small NW. The inherent chemical inhomogeneity also facilitates the concurrent activation of separated partial dislocation, HCP phase transformation, recurrence of reversible FCC-HCP phase transition, and zigzag-like nanotwin generation. Thus, the deformation mechanism of alloys should consider both the size effect and RSS-induced inherent chemical inhomogeneity, and this should be unique for RSS alloys.”.

Comment 4. For FCC lattice, stacking faults normally appear as two-layer HCP atoms. If there are several SFs stacking together, how can they be effectively distinguished from the FCC-HCP phase transition? Are there any possibilities that the phase transition mentioned in the article is just a pile-up of SFs.

Response: Thanks for the reviewer's comment. There are two paths for the FCC-HCP phase transition. The first path is the FCC-HCP phase transition via partial dislocation emission on alternating close-packed (111) planes, i.e., SFs that are one atomic layer away from each other, which causes the stacking sequence to change from “ABCABC” in an FCC lattice to “ABABAB.” The second is the Bain strain path, in which the FCC-HCP phase transition proceeds via “principal axis” straining without dislocation activities. Our in situ atomic-scale observations showed that the FCC-HCP phase transition in RSS AuCu NWs proceeds via partial dislocation emission on alternating close-packed (111) planes, which can also be seen as the pile-up of SFs (as the reviewer said).

In response the reviewer’s comments, we have added a sentence in the revised manuscript, as follows:

“This HCP lattice was generated via gliding partial dislocations by a single atomic layer away from SF3, and the continuous generation of SFs a single atomic layer away from the HCP phase boundary was repeated during straining, which caused the growth of the HCP phase, as shown in Fig. 5f. Our results provide direct evidence that the FCC-HCP phase transition occurs via partial dislocation emission on alternating close-packed (111) planes, i.e., SFs that are one atomic layer away from each other, which differs from the Bain strain path that proceeds via “principal axis” straining [75].”

Comment 5. MD simulations were applied to clarify the difference in between the deformation mechanisms of the RSS alloys and pure Au. However, it is not exact the same process as experiment (pure shear vs. uniaxial stress). It might be more convincing if the author can mimic similar loading process (uniaxial stress) in MD simulations.

Response: Thanks for the reviewer’s suggestion. In response, we have simulated the uniaxial tensile deformation in AuCu NW, as shown in Fig. S13. The diameter of the NW is 6 nm and the tensile direction is [111] with a strain rate of $\dot{\epsilon}=0.0001 \text{ ps}^{-1}$. During the deformation, we observed the presence of nanometer-thick deformation twins and HCP phase that is similar to the pure shear simulation results presented in the main text. Also, no evidence of broadening of the existing nanotwins and HCP phase with increasing tensile strain was found, a phenomenon that is contrary to the continuous twin propagation in pure metals. Thus, our MD simulation results were generally consistent with the experimental observations.

Also, it should be noted that the MD simulations presented in Fig. S13 did not perfectly reconstruct the experimental conditions, so their deformation behavior was not exactly the same. For example, the experimental tensile direction deviated a bit from the [111] direction. Also, a combination of shear deformation is also coupled in experiments, whereas MD simulation mimics only the pure uniaxial tension. Moreover, other factors, such as the sample morphology and strain rate, are also different.

In response the reviewer’s comments, we have added this information in the revised manuscript, as follows:

“We also simulated the uniaxial tensile deformation in AuCu NW (Fig. S13), and the results agree with the pure shear simulation results and our experimental observation.”

Fig. S13 Nanostructure features of four AuCu nanowire samples under uniaxial tensile deformation. The atoms are colored according to common neighbor analysis: green, red, and white atoms represent face-centered cubic (FCC), hexagonal close-packed HCP), and other structures, respectively. Nanotwins are indicated by yellow shades. During the deformation, we can observe nanometer-thick deformation twins and HCP phase that is similar to pure shear simulation results in the main text.

Also, no evidence of broadening of the existing nanotwins and HCP phase with increasing tensile strain was found, a phenomenon that is contrary to the continuous twin propagation in pure metals. Thus, our molecular dynamics (MD) simulation results are generally consistent with the experimental observation. This MD simulation did not perfectly reconstruct the experimental conditions, so the deformation behavior is not exactly the same.

Reference

- [1] Zhou, X. W., Johnson, R. A. & Wadley H. N. G. Misfit-energy-increasing dislocations in vapor-deposited CoFe/NiFe multilayers. *Phys. Rev. B* **69**, 144113 (2004).

REVIEWERS' COMMENTS

Reviewer #1 (Remarks to the Author):

The authors have satisfactorily addressed all my comments. I recommend accepting the manuscript in its current state.

Reviewer #2 (Remarks to the Author):

The authors have adequately addressed all of my concerns. The manuscript in the current form can be published.

Reviewer #3 (Remarks to the Author):

The authors have addressed all the comments in details and produced an enhanced version of the manuscript. Thus, it is now recommended for publication.